# Circulating Tumor Cells in Head and Neck Squamous-Cell Carcinoma Exhibit Distinct Properties Based on Targeted Epithelial-Related Markers

**DOI:** 10.3390/cimb47040240

**Published:** 2025-03-29

**Authors:** Kazuaki Chikamatsu, Hideyuki Takahashi, Hiroe Tada, Miho Uchida, Shota Ida, Yuichi Tomidokoro, Masaomi Motegi

**Affiliations:** Department of Otolaryngology-Head and Neck Surgery, Gunma University Graduate School of Medicine, Maebashi 371-8511, Gunma, Japan; htakahas@gunma-u.ac.jp (H.T.); tada-hiroe@gunma-u.ac.jp (H.T.); umiho@gunma-u.ac.jp (M.U.); ida.shota6666@gmail.com (S.I.); ytomidokoro@gunma-u.ac.jp (Y.T.); m_motegi@gunma-u.ac.jp (M.M.)

**Keywords:** circulating tumor cell, head and neck squamous-cell carcinoma, epithelial cell adhesion molecule, epidermal growth factor receptor, c-MET, liquid biopsy

## Abstract

The detection of circulating tumor cells (CTCs) using immunoaffinity-based methods often relies on epithelial-related markers, which may bias the selection of CTCs and limit the biological information obtained, depending on the targeted antigens. Herein, we compared the molecular profiles and clinical significance of CTCs based on the expression of epithelial-related markers (*EPCAM*, *EGFR*, and *MET*) in patients with head and neck squamous-cell carcinoma (HNSCC). CTCs were detected using density gradient separation and CD45-negative selection, followed by quantitative PCR for epithelial-related marker expression. Expression profiles of epithelial–mesenchymal transition (EMT)-related (*VIM*, *CDH1*, *CDH2*, *SNAI1*, *ZEB1*, *ZEB2*, and *TWIST1*) and immune-regulatory (*CD274* and *PDCD1LG2*) genes were compared. Moreover, the association between marker expression and clinical factors was analyzed. Among the 60 patients with CTCs, 48 (80.0%), 20 (33.3%), and 31 (51.7%) were positive for *EPCAM*, *EGFR*, and *MET*, respectively. A significant correlation was observed between CTCs expressing *EPCAM* and *EGFR*. CTCs expressing distinct markers showed differing EMT-related and immune-regulatory gene expression. *EPCAM*+ CTCs were associated with advanced-stage disease, while *EGFR*+ CTCs were correlated with locoregional relapse and shorter progression-free survival (*p* = 0.007; hazard ratio = 3.254). Patients with *EPCAM*/*EGFR* double-positive CTCs had the poorest prognosis. These findings emphasize the importance of marker selection in liquid biopsy technologies and highlight the need for improved detection methods and the further investigation of CTC biology.

## 1. Introduction

Growing evidence indicates that circulating tumor cells (CTCs) shed from primary tumors play an important role as “seeds” in the development of locoregional recurrence or distant metastases. Therefore, CTC analysis has been recognized as a predictive and prognostic marker in liquid biopsy for various cancers [1,2,3]. Among the various CTC detection methods, the most widely used method is immunoaffinity-based detection, such as immunomagnetic bead separation and non-magnetic antigen selection [4,5,6]. Given that carcinomas are of epithelial origin, the expression of epithelial-related markers is crucial for the isolation, enrichment, and identification of CTCs. Epithelial cell adhesion molecule (EpCAM) has been identified as the most common epithelial lineage [7], although it may cause the loss of some CTCs owing to the heterogeneity of targeted antigens on CTCs. In particular, CTCs undergoing epithelial–mesenchymal transition (EMT) are known to result in the loss of EpCAM, as well as acquire malignant potential, such as metastatic ability and treatment resistance [8], which may obscure the clinical significance of CTCs in liquid biopsy. Moreover, recent studies have demonstrated that CTCs co-expressing epithelial and mesenchymal phenotypes, representing a partial EMT state, are associated with increased invasiveness, immune evasion, and metastatic potential [9,10,11]. Mesenchymal markers such as vimentin and twist have been utilized to characterize these CTC subpopulations, which may be overlooked by epithelial marker-based detection methods.

Physical property-based methods and multiple epithelium-related markers have been developed and tested to overcome these limitations. In our previous study, we demonstrated that targeting three epithelial-related markers, EpCAM, epidermal growth factor receptor (EGFR), and MET, increased the CTC detection rate in patients with head and neck squamous-cell carcinoma (HNSCC) [12]. However, the characteristics and clinical significance of CTCs expressing these target epithelial markers remain unclear. In the present study, we compared the EMT status and immune regulatory ability of CTCs, distinguished by the expression of targeted epithelial-related markers in HNSCC, and evaluated the clinical significance of CTCs expressing each epithelial marker. 

## 2. Materials and Methods

### 2.1. Patients and Sample Collection

A total of 97 peripheral blood samples were collected, including 23 newly enrolled patients with HNSCC and 74 patients analyzed in a prior study [12]. All patients received similar treatments and shared comparable clinical characteristics, ensuring homogeneity between the two groups. None of the patients received any treatment prior to blood sample collection. Clinical information such as age, sex, primary tumor site, T classification, N classification, M classification, overall stage, human papillomavirus (HPV) (p16) status, and recurrence history was obtained from medical records.

### 2.2. CTC Detection and Gene Expression Analysis

CTC detection was performed as described previously [12,13]. Peripheral blood mononuclear cells (PBMCs) were isolated by density gradient centrifugation and treated with red blood cell lysis buffer (Roche Diagnostics GmbH, Mannheim, Germany). CD45+ cells were depleted using a human CD45 depletion cocktail and magnetic particles (EasySep Human CD45 Depletion Kit II; STEMCELL Technologies, Vancouver, BC, Canada). Total RNA was extracted using the RNeasy Micro Kit (QIAGEN, Hilden, Germany) according to the manufacturer’s instructions. cDNA was synthesized using the QuantiTect Reverse Transcription Kit (QIAGEN), followed by pre-amplification using the TaqMan™ PreAmp Master Mix Kit (Thermo Fisher Scientific, Waltham, MA, USA). Gene expression was analyzed using quantitative (real-time) PCR (Thermo Fisher Scientific). The specificity and sensitivity of the epithelial-related marker detection were previously validated by spiking experiments using an HNSCC cell line, demonstrating a correlation between the number of spiked tumor cells and Ct values in RT-PCR [14]. In addition, the expression of epithelial-related markers was also assessed in PBMCs from 40 healthy donors, with no detectable expression observed in any sample [12,14].

CTC positivity was determined based on the detection of at least one of three epithelial-related genes (*EPCAM*, *EGFR*, and *MET*). Among the 97 patients, CTCs were detected in 60 (45 from a previous study and 15 from new cases). The 60 CTC-positive samples were further analyzed to detect the expression of EMT-related genes, including *VIM*, *CDH1*, *CDH2*, *SNAI1*, *ZEB1*, *ZEB2*, and *TWIST1*. For the 45 previously studied CTC-positive samples, EMT-related gene expression analysis was performed previously. Therefore, only the remaining 15 CTC-positive samples were analyzed for EMT-related gene expression. Additionally, we investigated the expression of immune-regulatory genes *CD274* and *PDCD1LG2* for all 60 CTC-positive cases to evaluate the immune-regulatory properties of CTCs. *ACTB* was used as a control for normalization. All primers were purchased from Thermo Fisher Scientific (TaqMan Gene Expression Assays; Appendix A).

### 2.3. Statistical Analysis

Statistical analyses were performed using GraphPad Prism version 8.0 (GraphPad Software, San Diego, CA, USA) and SPSS Statistics for Windows, Version 29.0.1.0 (IBM Corp., Armonk, NY, USA). The Mann–Whitney U test was used to determine significant differences in continuous variables. The chi-squared test for independence and Fisher’s exact test were used to compare categorical variables. Survival curves were generated using the Kaplan–Meier method and compared with the log-rank test. Multivariate analysis was performed using the Cox proportional hazards model. Two-sided *p*-values < 0.05 were considered statistically significant.

## 3. Results

### 3.1. Characteristics of Patients with CTCs with HNSCC and Epithelial-Related Gene Expression in CTCs

Sixty patients with CTC-positive HNSCC were enrolled in this study. The patients’ clinical characteristics are presented in Table 1. *EPCAM*-positive, *EGFR*-positive, and *MET*-positive CTCs were detected in 48 (80.0%), 20 (33.3%), and 31 (51.7%) patients, respectively. Thereafter, we investigated the correlation between positivity rates of CTCs expressing *EPCAM*, *EGFR*, and *MET*. As shown in Table 2, a significant correlation was detected between CTCs expressing *EPCAM* and *EGFR* but not between those expressing other epithelial-related markers.

### 3.2. Comparison of Molecular Characteristics of CTCs Based on Epithelial Marker Expression

To evaluate the EMT status and immune regulatory ability of CTCs, seven EMT-related genes (*VIM*, *CDH1*, *CDH2*, *SNAI1*, *ZEB1*, *ZEB2*, and *TWIST1*) and two immune regulatory genes (*CD274* and *PDCD1LG2*) were analyzed and compared based on the expression of each epithelial marker in CTCs. As shown in Table 3, the expression levels of some genes in CTCs differed significantly depending on the expression of epithelial marker genes. *EPCAM*-positive CTCs showed significantly higher *CDH2* expression and lower expression of *CDH1* and *TWIST1* than *EPCAM*-negative CTCs. *EGFR*-positive CTCs exhibited significantly higher *CDH1* and *TWIST1* expression and lower *CDH2* and *ZEB1* expression than *EGFR*-negative CTCs. *MET*-positive CTCs showed significantly higher expression of *VIM* and *ZEB1* than *MET*-negative CTCs. Regarding immune regulatory molecules, *MET*-positive CTCs showed significantly higher expression of both *CD274* and *PDCD1LG2* than *MET*-negative CTCs.

### 3.3. Relationship Between the Expression of Epithelial-Related Markers in CTCs and Clinical Factors, Including Prognosis

We explored the relationship between CTCs and clinical factors and found that *EPCAM*-positive CTCs were present in patients with advanced-stage disease. Patients with *EGFR*-positive CTCs had significant locoregional relapse compared with those with *EGFR*-negative CTCs (Table 4). Notably, patients with *EGFR*-positive CTCs had significantly poorer progression-free survival (PFS) than those with *EGFR*-negative CTCs (*p* = 0.009) (Figure 1A). Meanwhile, regarding overall survival (OS), a trend toward inferior OS was observed for all types of CTCs, although the difference did not reach significance (Figure 1B). Interestingly, patients with *EPCAM*/*EGFR* double-positive CTCs exhibited a worse prognosis than those who lacked these characteristics (Figure 1A,B). Furthermore, multivariate analysis revealed that the presence of *EGFR*-positive CTCs was independently associated with worse PFS (*p* = 0.007; hazard ratio, 3.254; 95% confidence interval, 1.375–7.701) (Figure 2).

## 4. Discussion

In the present study, we demonstrated the differences in the expression of epithelial markers commonly used to identify CTCs. The three epithelial markers examined in this study are epithelial-specific genes, and contaminated white blood cell-derived materials are highly unlikely to interfere with gene expression in CTCs [15]; these epithelial-specific genes are well-known useful markers for CTC identification and enumeration [7,16,17,18].

EpCAM is expressed in most cancers and reportedly exerts diverse biological functions [19]. For instance, EpCAM is involved in cell-to-cell adhesion, cell proliferation, and cancer stemness, playing a role in tumor progression and treatment resistance. In tongue cancer, EpCAM expression was found to be substantially associated with tumor size, regional lymph node metastasis, histological differentiation, and invasion patterns [20]. Consistently, Murakami et al. demonstrated that intense EpCAM expression was an independent adverse prognostic factor in patients with HNSCC treated with primary radiation therapy [21]. In contrast, the loss or reduced EpCAM expression contributes to the induction of EMT, allowing tumor cells to acquire migration, invasion, and metastatic abilities. Therefore, low EpCAM expression was identified as a negative prognostic marker [22,23,24]. Additionally, the concept of partial-EMT state—an intermediary stage in the EMT plasticity between the epithelial and mesenchymal states—is widely recognized and plays a critical role in enabling tumor cells to exert various aggressive biological features [25,26]. Thus, the presence or absence of EpCAM expression in tumor cells, including CTCs, cannot be linked to the malignant traits of tumor cells [27].

EGFR plays a critical role in the pathogenesis of HNSCC and is frequently overexpressed in HNSCC [28]. EGFR activation also leads to proliferation, angiogenesis, invasion, the inhibition of apoptosis, and metastasis in tumor cells via the activation of various signaling pathways [29]. EGFR levels are known to be elevated in advanced-stage and poorly differentiated tumors, and its overexpression has been associated with worse prognosis [30]. In the present study, the expression of *EGFR* in CTCs was associated with locoregional relapse, and the presence of *EGFR*-positive CTCs was significantly related to shorter PFS in multivariate analyses, suggesting that *EGFR* expression in CTCs also reflects the malignant phenotype of tumor cells, similar to that in tumor tissue. In contrast, a significant positive correlation was detected between *EpCAM* and *EGFR* expression in CTCs. A close association has been reported between EpCAM and EGFR expression. For instance, in epithelial ovarian cancer, a positive correlation has been detected between EpCAM and EGFR expression in tumor tissues [31]. Interestingly, Pan et al. demonstrated that in head and neck cancer, the extracellular domain of EpCAM can bind to EGFR and induce EGFR-dependent proliferation [32]. These findings suggest that the presence of CTCs co-expressing *EpCAM* and *EGFR* contributes to a worse prognosis.

Regarding *MET* expression in CTCs, *MET*-positive CTCs exhibited a significantly higher expression of immune regulatory molecules *CD274* and *PDCD1LG2* in CTCs. In patients with non-small-cell lung cancer, Domènech et al. demonstrated that those with MET amplification had a higher proportion of programmed death-ligand 1 (PD-L1) expression and overexpression than those with non-amplified MET [33]. Moreover, MET inhibition and knockdown were found to downregulate co-inhibitory molecules such as PD-L1 and PD-L2. Conversely, MET activation can lead to increased PD-L1 expression [34], suggesting that MET overexpression/activation in tumors contributes to immune evasion by upregulating co-inhibitory molecules, thereby impairing immune surveillance. In our previous study, we demonstrated that patients with *MET*-positive CTCs have a poorer prognosis than those with *MET*-negative CTCs upon examining patients with recurrent/metastatic HNSCC treated with nivolumab [10]. These findings highlight the potential of combining MET-targeted therapy with immune checkpoint inhibitors for MET-dependent cancers [35].

The limitations of our study include the small sample size and constraints associated with CTC detection. In particular, with regard to the latter, it is important to note that various detection techniques are available, and the detection rate and/or the target positivity rate may vary depending on the method employed. Moreover, our detection strategy focused exclusively on epithelial-related markers (*EPCAM*, *EGFR*, and *MET*) and did not account for CTC subpopulations exhibiting mesenchymal or hybrid epithelial–mesenchymal phenotypes. A further investigation of such CTCs will be important, given their potential clinical relevance in tumor progression, metastasis, and treatment resistance. Addressing this issue is critical to advancing biomarker development through liquid biopsies using CTCs.

## 5. Conclusions

Various epithelial or cancer-specific markers are currently utilized for identifying CTCs; however, the selection of markers used for detection not only affects the biological properties of CTCs but also influences the assessment of clinical outcomes. To develop liquid biopsy techniques that can more accurately evaluate a patient’s oncological condition and provide more sensitive prognostic predictions, it is essential to improve CTC detection methods and further elucidate the characteristics of CTCs within the blood microenvironment.

## Figures and Tables

**Figure 1 cimb-47-00240-f001:**
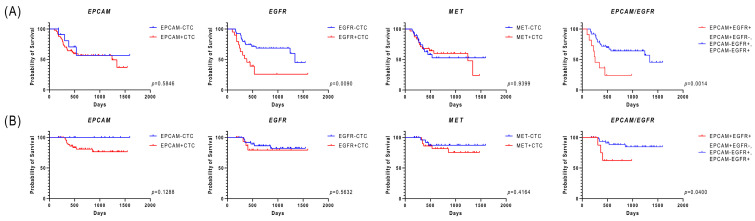
Kaplan–Meier survival analysis of CTC-positive patients with HNSCC. (**A**) Progression-free survival based on the expression of each epithelial-related gene. (**B**) Overall survival based on the expression of each epithelial-related gene. CTC, circulating tumor cell; HNSCC, head and neck squamous-cell carcinoma.

**Figure 2 cimb-47-00240-f002:**
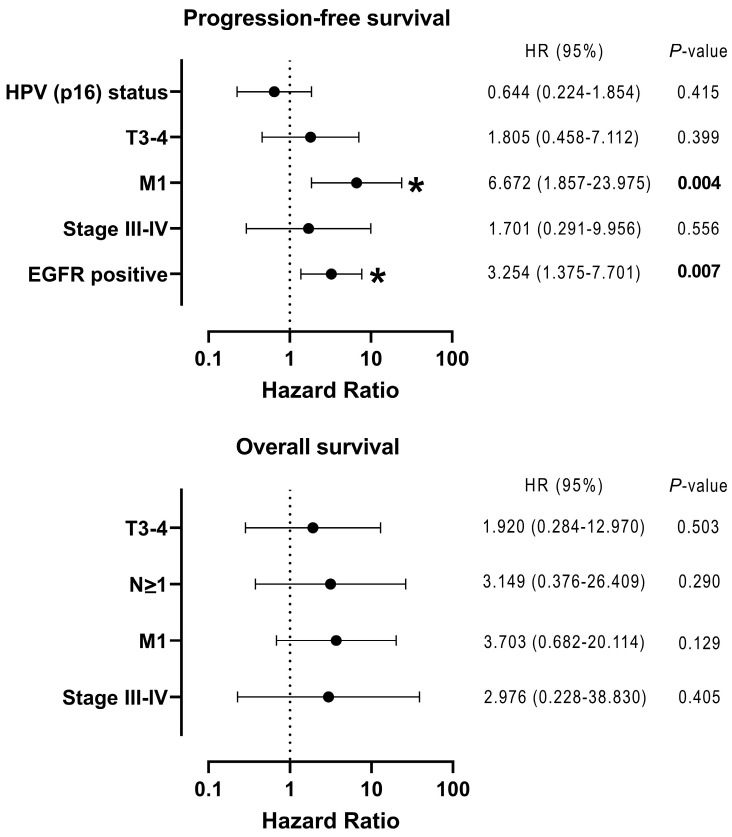
Forest plot based on multivariate Cox proportional hazards regression analysis in patients with head and neck squamous-cell carcinoma. HPV, human papillomavirus. Asterisk and bold values indicate statistical significance (*p* < 0.05).

**Table 1 cimb-47-00240-t001:** Clinical characteristics of 60 patients with CTCs.

Clinical Variable		Number of Patients (%)
Age: median 69 years old	≤69	31 (51.7)
	>69	29 (48.3)
Sex	Male	55 (91.7)
	Female	5 (8.3)
Primary sites	Oral cavity	2 (3.3)
	Larynx	14 (23.3)
	Oropharynx	28 (46.7)
	Hypopharynx	16 (26.7)
HPV (p16) status	Negative	36 (60.0)
	Positive	24 (40.0)
T classification	1–2	27 (45.0)
	3–4	33 (55.0)
N classification	0	20 (33.3)
	≥1	40 (66.7)
M classification	0	56 (93.3)
	1	4 (6.7)
Stage	I–II	24 (40.0)
	III–IV	36 (60.0)
*EPCAM*	Negative	12 (20.0)
	Positive	48 (80.0)
*EGFR*	Negative	40 (66.7)
	Positive	20 (33.3)
*MET*	Negative	29 (48.3)
	Positive	31 (51.7)

CTCs, circulating tumor cells; HPV, human papillomavirus.

**Table 2 cimb-47-00240-t002:** Correlations of epithelial-related marker expression on CTCs.

		*EGFR*		*p*-Value
		Negative (n = 40)	Positive (n = 20)	
*EPCAM*	Negative (n = 12)	3	9	**0.0013**
	Positive (n = 48)	37	11	
		*MET*		*p*-Value
		Negative (n = 29)	Positive (n = 31)	
*EPCAM*	Negative (n = 12)	7	5	0.5269
	Positive (n = 48)	22	26	
		*MET*		*p*-Value
		Negative (n = 29)	Positive (n = 31)	
*EGFR*	Negative (n = 40)	16	24	0.1004
	Positive (n = 20)	13	7	

CTCs, circulating tumor cells. Bold values indicate statistically significant differences (*p* < 0.05).

**Table 3 cimb-47-00240-t003:** Comparison of gene expression profiles in CTCs classified by epithelial marker expression.

		*VIM*	*p*-Value	*CDH1*	*p*-Value	*CDH2*	*p*-Value	*SNAI1*	*p*-Value	*ZEB1*	*p*-Value	*ZEB2*	*p*-Value	*TWIST1*	*p*-Value	*CD274*	*p*-Value	*PDCD1LG2*	*p*-Value
*EPCAM*	Negative	35.97 (34.39–36.85)	0.877	26.11 (25.92–27.73)	**0.025**	21.10 (18.19–22.79)	**0.017**	25.49 (24.33–27.05)	0.877	28.14 (26.98–29.00)	0.448	32.97 (31.99–33.11)	0.891	24.07 (23.37–25.13)	**0.002**	25.11 (24.98–26.74)	0.143	23.93 (23.16–25.42)	0.222
	Positive	36.01 (35.03–36.48)		25.18 (24.95–26.07)		23.00 (22.05–24.01)		25.50 (24.02–26.07)		28.51 (27.98–29.04)		32.99 (32.00–33.05)		23.09 (22.08–24.04)		26.02 (25.11–26.27)		24.34 (23.98–25.18)	
*EGFR*	Negative	36.02 (35.05–36.82)	0.171	25.13 (24.96–26.02)	**0.010**	23.03 (22.19–24.05)	**0.001**	25.90 (24.03–26.73)	0.658	28.98 (28.00–29.06)	**0.012**	32.99 (32.01–33.07)	0.182	23.08 (21.96–24.02)	**0.013**	26.02 (25.14–26.96)	0.054	24.64 (23.98–25.28)	0.127
	Positive	35.97 (34.32–36.04)		26.15 (25.24–27.02)		22.03 (19.19–22.99)		25.02 (24.33–26.06)		28.04 (26.98–28.77)		32.95 (31.98–33.01)		23.97 (23.09–25.04)		25.38 (25.03–26.05)		24.10 (23.12–25.07)	
*MET*	Negative	35.04 (34.51–36.32)	**0.002**	25.06 (24.89–26.10)	0.121	22.84 (20.10–23.07)	0.058	25.94 (23.65–26.05)	0.436	28.02 (26.98–29.02)	**0.038**	32.97 (32.00–33.04)	0.354	23.14 (22.47–25.04)	0.803	25.22 (25.03–26.08)	**0.023**	24.06 (23.18–24.64)	**<0.001**
	Positive	36.04 (36.00–36.89)		25.98 (25.02–27.00)		23.03 (22.05–24.05)		25.14 (24.32–27.02)		28.95 (28.02–29.04)		33.00 (32.01–33.07)		23.17 (22.90–24.05)		26.04 (25.53–26.97)		25.08 (24.16–26.03)	

CTCs, circulating tumor cells; Gene expression levels are presented as 40-delta Ct values. Data are expressed as the median with interquartile range (Q1–Q3). Bold values indicate statistically significant differences (*p* < 0.05).

**Table 4 cimb-47-00240-t004:** Correlation between the expression of epithelial-related markers on CTCs and clinical factors.

		Number	*EPCAM*			*EGFR*			*MET*		
			Negative	Positive	*p*-Value	Negative	Positive	*p*-Value	Negative	Positive	*p*-Value
			(n = 12)	(n = 48)		(n = 40)	(n = 20)		(n = 29)	(n = 31)	
HPV (p16) status	Negative	36	6	30	0.4292	26	10	0.2636	16	20	0.4603
	Positive	24	6	18		14	10		13	11	
T classification	1–2	27	8	19	0.1139	19	8	0.7836	14	13	0.7955
	3–4	33	4	29		21	12		15	18	
N classification	0	20	6	14	0.1893	15	5	0.3950	8	12	0.4194
	>1	40	6	34		25	15		21	19	
M classification	0	56	12	44	0.5744	37	19	>0.9999	27	29	>0.9999
	1	4	0	4		3	1		2	2	
Stage	I–II	24	8	16	**0.0498**	16	8	>0.9999	11	13	0.7969
	III–IV	36	4	32		24	12		18	18	
Relapse	Negative	34	8	26	0.5258	26	8	0.0975	17	17	0.7997
	Positive	26	4	22		14	12		12	14	
Locoregional relapse	Negative	43	9	34	>0.9999	33	10	**0.0143**	21	22	>0.9999
	Positive	17	3	14		7	10		8	9	
Distant metastasis	Negative	50	11	39	0.6697	32	18	0.4714	25	25	0.7324
	Positive	10	1	9		8	2		4	6	

CTCs, circulating tumor cells; HPV, human papillomavirus. Bold values indicate statistically significant differences (*p* < 0.05).

## Data Availability

Data are contained within the article or Appendix A.

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
