# Peer review of "Circulating Tumor Cells in Head and Neck Squamous-Cell Carcinoma Exhibit Distinct Properties Based on Targeted Epithelial-Related Markers"

_cimb, 2025, doi:10.3390/cimb47040240_

Round 1

Reviewer 1 Report

Comments and Suggestions for Authors

The paper is inetresting, but requires major revision before acceptance for publication.

In its present form it has three parts, and need changes:

1. The first part is a classification problem: do the epithelial-related markers (EPCAM, EGFR, and MET) have influence on the molecular profiles and clinical significance of CTCs.

2. Relationship between epithelial-related markers and clinical features

3. Survival analysis.

It think that authors should concentrate on descriptive statistics and survival analysis, because the importance of factors is reflected in its influence on survival, and consequently in its influence on therapy.

Table 1 and 2 should be retained but Figures 1 and 2 are not needed. Figure 1 is complicated to understand, and should be replaced with descriptive tables for all patients, and positive and negative epithelial-related markers. Figure 2 contains correlations of very low value (0.04 to 0.41), and information in this table has no practical value.

Table 3 and Figure 3 should be replaced with better version of of survival analysis. Univariate Cox analysis is not needed. Complete multivariate Cox with hazard ratios plot should be presented. Recursive partitioning (survival trees) is even better than Cox regression because this method has ability to subdivide patients into risk subgroups, and it is very easy to interpret. A fresh reference about Cox and survival trees is a book:

Prabhanjan Narayanachar Tattar, H. J. Vaman, Survival analysis. CRC Press, Boca Raton, 2023. Chapters 5 and 8 are very valuable.

Reviewer 2 Report

Comments and Suggestions for Authors

In the presented manuscript, the authors describe a clinical relevant study of  CTC-associated epithelial biomarkers in HNSCC.  The authors could support that the expression of EpCAm, EGFR, and MET can be relevant for disease progression and clinical outcomes . 

In summary, the authors provided a well-designed study, which is relevant for the field of liquid biopsies. The presented results are mostly convincing, although some more detailed information could be given. Thus, I recommend acceptance of the study after minor revisions.

  1. Introduction is quite short and can give some more info about different markers relevant for liquid biopsies, eg also mesenchymal markers
  2. in general the relevance of mesenchymal markers is neglected in the study. It is known that CTCs can also exhibit an "intermediate" state of surface marker expression with epithelial and mesenchymal markers (such as cell-surface vimentin) which seem to be especially relevant for malignancy and metastasis. This fraction of CTC is not considered at all. Authors should at least comment on that.
  3. although there seem to be previous studies, the manuscript should give detailed info about the isolation method, maybe also presenting convincing data on the identity of the isolated cells (tumor-marker staining or else) since a negative depletion method was used with the risk of contamination.

Round 2

Reviewer 1 Report

Comments and Suggestions for Authors

The authors have addressed all objections and significantly improved the work, which is now more condensed and easier to follow.

An advice for future work- Authors have tried recursive partitioning, (party module in R). In my opinion rpart and rpart.plot modules in R programming language are better alternative in comparison to party.